# Optimization of Hexagonal Structure for Enhancing Heat Transfer in Storage System

**DOI:** 10.3390/ma16031207

**Published:** 2023-01-31

**Authors:** Natalia Raźny, Anna Dmitruk, Artur Nemś, Magdalena Nemś, Krzysztof Naplocha

**Affiliations:** 1Department of Lightweight Elements Engineering, Foundry and Automation, Faculty of Mechanical Engineering, Wrocław University of Science and Technology, Wybrzeże Wyspiańskiego 27, 50-370 Wrocław, Poland; 2Department of Thermodynamics and Renewable Energy Sources, Faculty of Mechanical and Power Engineering, Wrocław University of Science and Technology, Wybrzeże Wyspiańskiego 27, 50-370 Wrocław, Poland

**Keywords:** heat storage, heat transfer, PCM, spatial structures, honeycomb, metal structures

## Abstract

Thermal performance was tested during cycling work for latent heat storage systems based on KNO_3_ and NaNO_3_ (weight ratio 54:46). For heat transfer improvement, cast aluminum honeycomb-shaped structures were produced via 3D printing of polymer model and investment casting. Different wall thicknesses were tested at 1.2 mm and 1.6 mm. The obtained results were compared to working cycles of pure PCM bed. The use of enhancers is reported to improve the rate of charging and discharging of the deposit. In the next step, the structures were examined with numerical simulation performed with ANSYS Fluent software. The wall thicknesses taken into consideration were the following: 0.8, 1.2, 1.6, and 2.0 mm. An insert with a greater wall thickness allows for smaller dT/dt and better heat distribution in the vessel. The investment casting process enables the manufacturing of complex structures of custom shapes without porosity and contamination.

## 1. Introduction

Due to the currently growing worldwide demand for energy and increasing environmental constraints, the key issue appears to be developing new, safe, and sustainable energy sources, storage, and transport technologies. In this case, renewable energy sources (RES) such as solar, wind, or geothermal deserve special attention. An important aspect of thermal energy storage (TES) is the ability to gather large amounts of energy in relatively small volumes, which can be guaranteed by the high phase transition enthalpy, typical for some of the Phase Change Materials (PCMs). This feature allows them to accumulate large amounts of latent heat during a phase change (melting, evaporating) [1]. For solar energy storage, salt hydrates and their eutectic mixtures are the most commonly used PCMs. This choice is connected to the fact that the operating temperature of solar energy storage oscillates around 250 °C, in which organic materials can degrade. Nevertheless, the bottleneck limiting the wider application of PCMs in heat storage tanks is their low thermal conductivity, prolonging the charging period of such units.

Numerous materials and structures are being used to support the transport of heat in PCMs. Such, so-called heat transfer enhancers directly improve the thermal conductivity of the deposit, creating composite PCM. The following examples can be highlighted among the particles’ additives [2,3]: carbon-based fillers or nanofillers (e.g., expanded graphite, carbon nanotubes (CNTs)), 2D materials (e.g., graphene or MXenes), and metallic or ceramic powders (e.g., Al, Cu, SiO_2_, Al_2_O_3_, Fe_2_O_3_, ZnO). Variously shaped and manufactured by casting or plastic working, metal heat exchanging surfaces are also commonly applied as follows: tubes (plain, helical, or finned [2,3]), fins [4,5], metal foams (e.g., Cu, Al, Ni) [6] and biomimetic fractal-like (e.g., tree-, Y-, snowflake-shaped [7,8,9,10]), and pin-fin [11] or honeycomb structures [12]. Another method is micro- or macroencapsulation in organic or inorganic shells of superior thermal conductivity [5,6,13].

The main focus of this work will be put on the application of honeycomb structures in heat exchange, especially in PCM heat storage units. Lu et al. analyzed the factors such as wall thickness, cell size, and orientation and stated that the heat transfer characteristics of micro-cell Al honeycombs are alike the ones of open-cell Duocel^®^ foams [14]. Kong et al. tested and simulated numerically the thermal behavior of gradient honeycomb heat enhancer 3D-printed from 316 L stainless steel. Comparing the results with uniform structures the 14–17% overall thermal performance enhancement was observed [15]. In the work of Andreozzi et al., a ceramic cordierite honeycomb system with paraffin wax as PCM was subjected to numerical simulation for different pores per unit of length (PPU) values. It was found that higher PPU benefits with faster charging [16]. The influence of different geometry of cores (triangular, trapezoidal, rectangular, hexagonal, circular) on the thermal cycling of aluminum honeycomb structures with paraffin was researched by Duan and co-authors. It was stated that cells with a smaller geometrical factor (GF), such as triangular or quadrilateral, ensure the shortening of melting times of PCM, compared to a hexagonal structure. Moreover, the cell orientation appears to matter, as turning the hexagonal cell corner down resulted in a 9.9% reduction in melting duration [17]. Egolf et al. studied the possibility of the utilization of inorganic or organic (e.g., glass, polycarbonate (PC)) honeycombs in the construction of translucent solar collectors integrated with PCM (CaCl_2_6H_2_) heat storage modules to be used in buildings façades. Longitudinal honeycomb cells forming panels were considered with various degrees of filling with PCM (e.g., 33.3%, 66.6%—the rest of the hexagons were left empty) resulting in physical models, numerical simulation results, determination of honeycombs absorption properties, and solar impacts [18]. The aluminum honeycomb wallboard elaborated by Lai et al. was filled with microencapsulated PCM (mPCM, here paraffin–pure or with expanded graphite or ironwire, covered with polymer shell). A honeycomb sample (10 cm × 10 cm × 2.54 cm, core cell 8 mm) was tested experimentally in terms of thermal performance in PV panels providing sufficient heat conduction enhancement and unifying the interior temperatures of the module with no thermal stratification [19]. A subsequent study on mPCMs by similar authors confirmed the benefit of Al honeycomb working as a structural support and heat transfer channel facilitating thermal energy dissipation [20]. Another study reported an investigation of the effect of honeycomb core on latent heat storage (LHS) in PCM solar air heater (SAH). Three SAHs were compared, containing the following: a flat absorber plate, a heat storage panel with PCM (paraffin), and the one with PCM and a honeycomb core. The main findings were that, due to the utilization of aluminum honeycomb structure, the charge-discharge times were significantly reduced and the daytime temperature was increased [21]. A similar honeycomb was introduced also to the paraffin by Xie et al., resulting in a thermal conductivity increase up to 2.08 W/m·K [22]. Kant et al. considered n-octadecane as a PCM reinforced with aluminum honeycomb in numerical simulation investigating the effect of cells length, thickness, and inclination angle. The analyzed composite PCM exhibits an improved charging rate with the increasing volume of a metal insert, while the tilt influence is rather marginal. Nevertheless, it has to be taken into account that replacing the PCM volume with a honeycomb structure reduces the total energy stored, and therefore a compromising solution must be identified [12].

In this paper, the use of an efficient hexagonal cast aluminum structure to enhance heat transfer in the salt PCM deposit is proposed. In the experimental part, three arrangements are taken into consideration: the deposit filled only with salt PCM, and two deposits enhanced with honeycomb inserts with differing wall thickness of 1.2 mm and 1.6 mm. The inserts are designed by 3D modeling and cast by investment casting. The charging and discharging of the deposits supported by the structures are described. Thermal and physical phenomena occurring during operations are depicted. During the numerical part of the research, the influence of the wall thickness on the performance of the unit is studied. The analysis is performed for the following dimensions: 0.8 mm, 1.2 mm, 1.6 mm, and 2.0 mm.

## 2. Materials and Methods

### 2.1. Experimental Approach

Aluminum honeycomb inserts were manufactured by the means of the investment casting method, consisting of the design and execution of the model, molding with plaster, the burn-out cycle of the mold, and metal pouring under low pressure. Models were designed in Autodesk Inventor Professional 2018, then the Simplify3D software (Simplify3D 4.1.2, Simplify3D Inc., Cincinnati, OH, USA) was used to create the G-code file. Spatial models from polylactide (PLA) were produced with the use of HBOT 3D printer F300 (3D Printers, Wrocław, Poland). Thus, prepared patterns with attached wax gating systems after molding and initial hardening were burned out at the maximum temperature of 730 °C in the furnace. During this stage, the combustion and gasification of the polymer model occur and the plaster mold hardens and obtains its full strength. The evaporated pattern leaves the internal precise cavity of its shape in the plaster. Ceramic plaster Randolph Ransom-type R&R^®^ ARGENTUM™ (Ransom & Randolph, Maumee, OH, USA) (quartz < 50%, cristobalite < 50%, CaSO_4_ binder) was utilized. Mold, still at the elevated temperature, was mounted in autoclave and under the pressure of 0.04 MPa, the chosen aluminum alloy (AC 44200, Si-10.5%-13.5%; Fe-0.55%; Cu-0.05%; Mn-0.35%; Zn-0.10%; Ti-0.15%; Al-rest) was cast.

Cast inserts, varying in wall thicknesses (1.2 and 1.6 mm), were alternately located horizontally in the center of the isolated PCM-based accumulator, fully immersed in the chosen PCM—the eutectic composition of KNO_3_ and NaNO_3_ (Archem) with a weight ratio of 54:46 and a melting point of 222 °C. Both of the salts were purchased separately, weighed carefully, mixed, and melted together. Subsequently, the thusly prepared heat storage unit was subjected to multiple working cycles of charging and discharging. The temperature distribution in the unit was controlled with the use of three thermocouples located on the bottom (12 mm), in the center (50 mm), and on the top (87 mm) of the heat storage tank, and in the middle of the honeycomb cells. The scheme of the stand is presented in Figure 1. Charging and discharging cycles were conducted by the hot plate (400 °C) used as a heat source, in order to achieve the semi-directional heat flux in the height direction of the accumulator. The heating–cooling thermocouple placed in the cooling plate indicates the temperature of the heating source while charging the accumulator. When the deposit is completely charged, the salt is melted, and water cooling is activated. The temperature shown then by the thermocouple corresponds to the temperature of the water during cooling. Thus, it is possible to observe, by the changes in the water temperature, how the accumulator is discharged by transferring the heat from the PCM to the cooling plate and, then, to the water.

The deposit was heated up with the use of a heating plate (Hot Plate, Model SH-II-5B, heating power 1200 W, temperature accuracy ±1%). The experimental data, the temperature at different heights in the bed, were registered by K-type NiCr-NiAl thermocouples, class 1 (±1.5 °C) according to the PN-EN 60584 standard, collected, and recorded by Adam 4018 type adapter (American Advantech Corporation, Sunnyvale, CA, USA) with 8 channels, and the VisiDAQ program (VisiDAQ Professional Version 3.1, American Advantech Corporation, Sunnyvale, CA, USA) The loading step’s duration was dependent on the time needed to melt the entire deposit. The first trials established the performance of the accumulator filled only with the PCM material (11 h charging/discharging cycle), while the next concerned the composite PCM with the immersed aluminum honeycomb structures (5 h charging/discharging cycle).

### 2.2. Numerical Analysis—Geometry, Boundary, Initail Conditions, and Numercial Schemes

In the next step, the influence of the wall thickness of the insert on the melting process of PCM was analyzed. The heat flow in PCM can be described using standard equations [23,24]. Partial differential equations are used to solve the mass, momentum, and energy equations in complex systems [25]. For this purpose, the Finite Volume Method (FVM) was used [17], and the calculations were performed with the commercial Ansys^®^ Academic Research Fluent, 2021R1 (ANSYS, Inc., Canonsburg, PA, USA).

The meshes for all 12 domains shown in Figure 2 were carefully prepared to meet the high regime for phase change calculations. Their quality allowed for a high accuracy of calculations, which is described later. In addition, the mesh was densified near the walls to reduce the calculation error on the walls and in places of lower mesh quality, as shown in Table 1. The smallest thickness of adjacent elements was 0.05 mm.

In the numerical calculations, the geometry used in the experimental studies was used. The boundary conditions used in the simulations are shown in Figure 3.

The insert geometry used in the calculations is based on the experiments described in the previous chapters, and considers the following conditions:The vessel geometry has been simplified;The bottom surface temperature was 400 °C;The remaining walls were adiabatic;In order to speed up the calculations, the symmetry condition was used and only half of the geometry was calculated;PCM properties have been defined identically for each domain (2 ÷ 14).

The physicochemical properties of PCM and aluminum are presented in Table 2. Due to the small differences in specific heat and thermal conductivity of the liquid and solid PCM phases, the same values were set. The convective movement of the liquid phase of PCM is caused by the difference in density; therefore, it was dependent on the temperature and in the tested range, as described by Equation (1).
(1)$$\rho \left(T\right)=-0.8958T+2462.5$$

The geometry and numerical mesh were made in Ansys^®^ Academic Research ICEM CFD, 2021R1 (ANSYS, Inc., Canonsburg, PA, USA). Each of the closed spaces inside the insert was a separate domain filled with PCM. The PCM was also outside the insert. All 14 domains were discretized by using hexahedral elements. Additionally, on the outer surfaces, inflation layers were applied. The thickness of the first element adjacent to the wall was set to 0.07 mm. A laminar model for PCM was used. Calculations were performed in a transient mode with a time step of 0.05 s. The solution in each time-step was considered as converged if the residuals were less than 10^−3^, except for the energy equation, where it was 10^−6^. However, a maximum of 30 iterations per time-step was completed.

In order to determine the influence of the wall thickness of the insert on the rate of heat flow inside its cells, calculations were made for several wall thicknesses: 0.8, 1.2, 1.6, and 2.0 mm. A separate geometry was made for each thickness and a mesh was generated in accordance with the above rules. The boundary conditions were the same for each case.

## 3. Results and Discussion

This section presents the results of both research paths undertaken in the article. In the first part, the experimental observations are described and concluded, while in the second one, the discussion of the numerical simulation approach is depicted.

### 3.1. Experimental Results and Discussion

A detailed examination of the heating and cooling processes of the chamber filled with PCM and enhanced with prepared inserts was carried out and the best performing enhancer was selected. It should be taken into account that the duration of charging and discharging the storage unit was affected by error, due to heat losses in the environment, despite the use of insulation. Nevertheless, all measurements were conducted under the same conditions; therefore, the error can be treated as systematic and the results obtained can be compared.

#### 3.1.1. Investment Casting of Honeycomb Structures

Figure 4a presents the real cast insert placed in the horizontal position, in the same way as in the experiment, while in Figure 4b the designed model with the most important dimensions is presented. An appropriate selection of parameters based on the experience and observations allowed for obtaining a structure without visible surface defects.

#### 3.1.2. Experimental Heat Transfer Performance

Figure 5 shows the temperature variation over time for the thermocouples placed in the system. The units were heated for different periods due to the assumption that the heating is stopped approximately when the whole deposit is molten. In the case of the insert with a 1.2 mm thick wall, even in a significantly shorter time of charging, the deposit reaches a considerably higher temperature (285 °C) than for the pure salt deposit (235 °C).

It can be noticed that the use of the heat transfer enhancer has significantly accelerated the charging period and furthered the melting of the deposit. The influence is especially pronounced for the top thermocouple in both cases, and can be seen based on the different lengths of flattening in the graph. The plateau is related to the melting of the part of the salt PCM in the deposit, which is located outside the volume of the insert structure. The time required to reach the melting temperature is two times higher for the pure PCM deposit than for the one with the elaborated cast structure. Similar observations were reported in [21]. Abuşka et al. tested solar air heaters based on PCM with and without metal hexagonal enhancers placed in the deposit. Tests were driven for six different mass flow rates of air. They claimed that the use of honeycomb structures as heat enhancers can substantially shorten the time of charging and discharging the deposit. A structure applied in the deposit allowed the PCM temperature to increase by 8.8 °C (about 10%) in comparison to a deposit with only PCM.

In the case of crystallization, the long continuous line can be observed, especially in the case of the top (h = 87 mm) thermocouple for the salt deposit, while for the deposit with the insert crystallization, time is much lower. The discharging of the deposit can be successfully supported by the use of such structures.

The dashed reference lines in the graph indicate the time to reach the phase transition temperature in the bottom cell of the honeycomb (referred to as t_1_ in Table 3), and in the top cell (referred to as t_2_ in Table 3). Furthermore, lower source temperatures were recorded for the PCM deposit with the metallic structure than for the one without it. As mentioned in the section before, in all cases the target heating source temperature was set to 400 °C. At the beginning of each measurement, the source had a temperature of approx. 50 °C. The target temperature was never reached due to the absorption of the heat by the unit. In the case of pure salt units, the reached temperature of the source was higher than for the units with metal enhancers. It can be concluded that the insert is absorbing heat and transferring it into the deposit, so the source temperature slowly increases, but the phase transition process and temperature equalization in the heat storage tank are facilitated. The reduction in the surface temperature using encapsulated PCM was also highlighted in [19]. Lai and Hokoi presented a concept of a honeycomb wallboard filled with microencapsulated PCM, which can be used in building construction to control the temperature in the rooms. Paraffin was used as the PCM. The system was tested with a series of different heat fluxes. Results obtained suggest that the honeycomb structure placed inside the wall allows good heat conduction and the surface of the wall is characterized by the lowered temperature. Likewise, in the case of observations presented hereby, the temperature of the source while using the honeycomb-enhancing structure is lower than for the only-PCM unit since the heat is absorbed by the cast enhancer and transported to PCM located further in the deposit.

On the basis of the data presented in Table 3, one can point out the significant variation between the average derivative difference of each studied case. The time of gaining the melting temperature by the bottom thermocouple (h = 12 mm) t_1_ was similar in all three units. By analyzing the average Δt, it is observed that a cast enhancer with a wall of 1.2 mm thickness can shorten the time of charging the deposit twice, while the insert with a 1.6 mm thick wall can shorten the time of charging by over four times.

Figure 6 includes the time differential temperature plot for the measurements of the top thermocouple for the three variants tested. Positive values of the derivative indicate the absorption of heat energy by the system, while the negatives are responsible for releasing heat into the environment.

The first peak (A) is responsible for heat conduction through the insert. A sudden rise and subsequent drop are seen primarily for the dT/dt 1.6 mm curve, while a similar tendency but of lower values is observed for the dT/dt 1.2 mm dependence. In the case of an only-salt deposit, no such behavior was present as the temperature of the hotplate stabilizes. It could be concluded that the insert conducts heat throughout the deposit, while salt PCM by itself has an insulating nature and does not efficiently transport heat. Mihalka and Matiasovsky drew a parallel conclusion in [30], which they have proved by CFD simulations. Better conductivity was obtained for PCM enriched with metallic honeycomb enhancing structure than for the pure PCM deposit.

A slow decline in the value of the temperature derivative (B, characteristic for 1.2 mm) is connected to the phase change (melting) of PCM. The temperature equalizes in the whole deposit when the entire volume of PCM has undergone phase change. The stage ends with a sharp peak, indicating the start of the heating of the liquid phase. The peak falls quickly as the deposit is already heating evenly. Similar behavior can be observed earlier for the 1.6-thick-wall insert (at approximately 10,000 s), and, later, for the only-salt deposit (at approximately 17,000 s).

The following decrease in the derivative (C on the graph, for 1.2 mm insert) and its change to negative values are connected to switching off the heating plate and the start of water cooling. Water used in the experiment for cooling the system was at room temperature, approx. 17 °C. When the water reached the cooling system placed under the deposit, the system started to cool immediately. In the beginning, water was leaving the system in the form of steam, but then it quickly changed to liquid and started to decrease its temperature. The most rapid temperature decrease can be observed for the 1.6 mm-thick-wall honeycomb. Slightly varying behavior (less distinct decrease) was obtained for the 1.2 mm-thick-wall insert, while for pure salt deposit, only the stopping of heating was recorded. The derivative is oscillating around 0, which corresponds to keeping the constant temperature.

At the D point (1.2 mm), a crystallization process occurs. The derivative is close to 0 value, which indicates that no temperature changes are going on in the system as the PCM releases latent heat stored during solidification. Similar processes can be seen earlier for 1.6 mm honeycomb due to the facilitated heat transport.

Other peaks are observed for temperatures in the region of 130–150 °C (E, characteristic of the yellow curve). In this range, the transition from the rhombic to the trigonal solid phase (change of crystalline system) of KNO_3_ takes place [31]. The location of the peaks may be slightly changed due to the effect of the heating rate.

Figure 7 presents the temperature difference (ΔT) between the top (h = 87 mm) and bottom (h = 12 mm) thermocouple measurements over time for all tested configurations. The solely PCM deposit exhibits the highest temperature gradient, indicating limited heat flow. In the case of 1.6 and 1.2 mm-thick-wall honeycombs, the differences are significantly lower. The highest value for a pure salt deposit (175 °C) is three times bigger than the peak value for the insert with a 1.2 mm-thick wall (approx. 55 °C), and for 1.6 mm—approx. 30 °C. Those are the difference occurring during the charging of the accumulator. While discharging, the significant differences can be still observed—when the peak for pure salt deposit reaches approx. −73 °C, the lowest values for 1.2 mm is −26 °C, and the lowest for 1.6 mm is −14 °C.

The huge negative ΔT visible for only PCM deposit is strictly connected to the heat transfer rate in the bed. Salt mixture in solid state is characterized by low thermal conductivity (for KNO_3_-NaNO_3_ mixture 0.457 W/(m·K)); thus, even if the part of the deposit is already melted, the heat provided to the rest of the material is significantly lower than the amount enabling phase change. In case of enhanced structures, the heat is transported through the aluminum structure, a material with considerably higher thermal conductivity of 160 W/(m·K); therefore, the melting process, as well as that which is strictly connected to the material temperature, is more evenly distributed in the system.

As shown in Table 4, the temperature gradient for different deposit heights, at the point when the PCM in the bottom cell reaches its melting point (h = 12 mm), is reduced with the increase in the honeycomb’s wall thickness, resulting in improved heat dissipation. For all tested cases, the melting process starts at a similar temperature at all heights. However, a vast difference can be noticed between the temperature of top thermocouples in time t_1_, where pure PCM ΔT amounts to 165.1 °C, while for a 1.2 mm-thick-wall insert the temperature is 48 °C, almost four times less, and for a 1.6 mm-thick-wall insert—32.3 °C. At t_2_, the temperatures at all heights are close because the majority of the deposit is already melted.

The heat transfer enhancer absorbs energy from the heat source and transports it evenly through the deposit; thus, vast differences in the temperature at different heights of the accumulator were compensated. The thicker the walls of the cast structure, the more heat is received and transported through the store, and, in consequence, the shorter the time needed to melt the entire volume of PCM. The difference in the temperature value during the charging of the accumulator was also reported in [32]. The article aimed to present a numerical prediction of solid–liquid interface location and an understanding of temperature distribution in the deposit. Basing on a numerical one-dimensional model of the phase change material storage with a fin, they calculated and presented temperature distribution. The tank filled with paraffin PCM was heated by an aluminum fin structure. As the distance of the measurement point from the heat source increases, a lower PCM temperature is visible. The resulting gradient depends not only on the source temperature and the heat flux rate provided into the deposit, but on the thickness of the enhancing element as well. The increase in the diameter of the fin or, in our case, in the thickness of the wall, will result in lower gradient values and faster charging of the accumulator.

These conclusions are also verified in Figure 8, where the increase in the liquid phase of PCM over time is shown. Initially, there are no relevant differences between the cases considered; however, after melting of the bottom of the deposit, about 25% of the PCM volume, the process significantly accelerates for the tanks with metal inserts. The heat flux supported by metal enhancers moves more quickly through the deposit, which results in a notably shorter time required for a fully charged accumulator. In the case of a 1.2 mm-thick-wall insert, the entirely melted deposit was observed after 10,650 s (2.95 h), while for 1.6 mm, a complete transition to liquid phase takes place after 9190 s (1.5 h). In comparison with the only-salt deposit, in which full remelting occurs after 17,000 s (4.7 h), in which the differences are 1.75 h and 3.2 h, respectively.

### 3.2. Numerical Analysis Results and Discussion

Performing numerical calculations allowed the determination of the influence of the wall thickness of the insert on the change of PCM temperature and the heat transfer rate. Figure 9 shows the dependence of temperature in time at three heights corresponding to the height of thermocouples in experimental studies.

The time of the phase change is clearly visible for each height. Figure 9a shows a rapid temperature increase in the first minutes of the process. It is caused by the largest temperature gradient. This increase is not so visible for h = 50 mm (Figure 9b). This is due to the slow transport of heat in the vertical direction. The slowest temperature increase was recorded for the height was 87 mm. There is no clear difference between the temperature rise before and after the phase change at this height. This may be due to heat transport in the insert in multiple directions simultaneously. There is a noticeable influence of the insert thickness on the rate of temperature increase. It is worth noting that at the height of mm, PCM heats up the fastest for the insert with the smallest wall thickness. This is due to the slower transport of heat through the insert in the vertical direction. This causes the PCM at the bottom of the vessel to heat up faster (close to the heat source) and the phase change occurs faster. However, the temperature distribution changes with the height, and for h = 87 mm, the phase change occurs the fastest for an insert with a thickness of 2.0 mm. This proves as a better heat transport in the vertical direction than for an insert with a smaller wall thickness. At the end of the process, the differences are small as the PCM temperature tends to the same value, equal to the heat source temperature.

In order to better define the dynamics of the process, the temperature change in time dT/dt was determined, presented in Figure 10, for three different heights.

As shown in Figure 10, the heat transport rate is very high at the beginning of the process, causing large changes in temperature in time, mainly at the height of 12 mm, where the phase change begins the fastest. The temperature gradient is the highest among the three analyzed heights and exceeds 0.5 K/s. The rapidly increasing temperature over time is the result of a short distance from the heat source. Moreover, the highest temperature increase at h = 12 mm was obtained for the insert thickness of 0.8 mm. This may be due to slower heat transfer in the vertical direction. After this stage, there is a noticeable decrease in dynamics, especially in Figure 10b. It is caused by reaching the temperature at which the phase change takes place. At h = 12 mm, the dynamics decrease, but increase at other heights. It is worth noting that after the described phase change, the dT/dt values oscillate close to zero. This oscillation is also visible in Figure 10b, after the phase change. This may be due to the convective motion of the PCM and the sinking of the solid phase. This would correspond to the non-uniform temperature changes visible, especially in Figure 9a. The influence of the wall thickness at the height of 50 mm and 87 mm is visible, where the dynamics increase with increasing wall thickness. This confirms the observations described for h = 12 mm, for which the phase change occurs later, with the increase in wall thickness. This suggests an increased heat transfer along the insert to the top of the vessel. Figure 10b shows the effect of using a thicker insert. The phase change process is the slowest there for a wall thickness of 0.8 mm. However, for the other thicknesses, the phase change time is similar. Figure 10c shows the best effect of using inserts with a larger wall thickness. The phase change process at the height of h = 87 mm (Figure 10c) occurs the fastest for a wall thickness of 2.0 mm. It should be noted that the rate of temperature change in time does not increase linearly with increasing wall thickness. It is also visible in the maximum values of dT/dt, i.e., right after the PCM has been melted. For a wall thickness of 2.0 mm, there is the smallest increase in dT/dt during the phase change (shown in Figure 9c). For thicknesses 0.8, 1.2, 1.6, and 2.0, the maximum values were 0.94, 0.80, 0.52, and 0.45, respectively. This suggests a more even heat distribution through the insert. This also reduces the temperature gradient throughout the vessel, which is a desirable effect.

## 4. Conclusions

Based on the research and results presented, the conclusions can be drawn as follows:Additive manufacturing followed by investment casting allowed for obtaining customized complex metal structures;The use of heat enhancer can notably increase the rate of charging. In the case of studied systems, melting temperature was reached two times faster in case of the system enhanced with a 1.2 mm-thick structure in comparison to the pure deposit of salt PCM. The presence of the casting influences the behavior of the heat source.Due to the nonconductive properties of used salt mixture, the application of the insert with 1.2 mm walls improved the time of charging twice, while improving the time of 1.6 mm at a rate of four times. Additionally, the use of the thicker insert lowered the temperature gradient at the beginning of phase change in the deposit five times, compared to pure PCM.Even for salt mixture, phase change from rhombic to triagonal structure of KNO_3_ in the temperature range of 130–150 °C is observed in the form of flattering the temperature-in-time derivative plot.According to the numerical analysis, which confirmed the beneficial effect of using inserts with higher wall thicknesses, it can be concluded that the higher the thickness of the enhancer wall, the better the distribution of heat throughout the vessel volume is observed.The main innovation presented in the article is a manufacturing method which allows new, thin-walled, fully customized complex structures, with the possibility of rapid modifications, to be fabricated. The elements produced via investment casting method are characterized by a high precision of execution, repeatability, and extensively developed surface area, allowing good contact between PCM and the material of the enhancer. Before that, the enhancers were commonly produced by stainless steel by the means of extrusion, which was a huge limitation when it comes to the use of complex shapes. Moreover, the use of stainless steel in molten salt environment carries a high risk of corrosion of the system, so the service life of the system will be in danger of being drastically reduced. The use of a customized aluminum enhancer reduces the chances of such phenomena and allows the insert to be optimally adapted to the case study. In addition, the developed method allows the production of any metal shape quickly, and is much cheaper than, e.g., selective laser melting (SLM).

## Figures and Tables

**Figure 1 materials-16-01207-f001:**
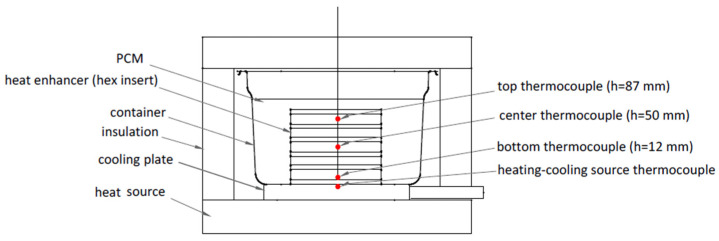
Test stand scheme: the thermocouples are placed in the center of the cells, in the middle of the structure.

**Figure 2 materials-16-01207-f002:**
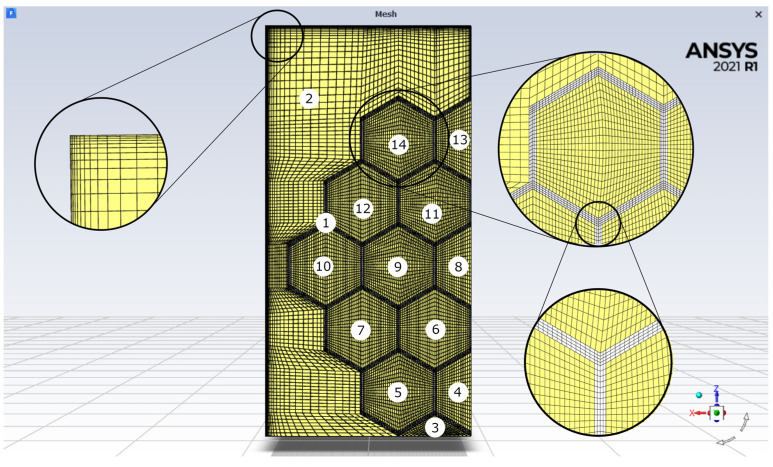
Mesh with details used during numerical calculations. Images used courtesy of ANSYS, Inc.

**Figure 3 materials-16-01207-f003:**
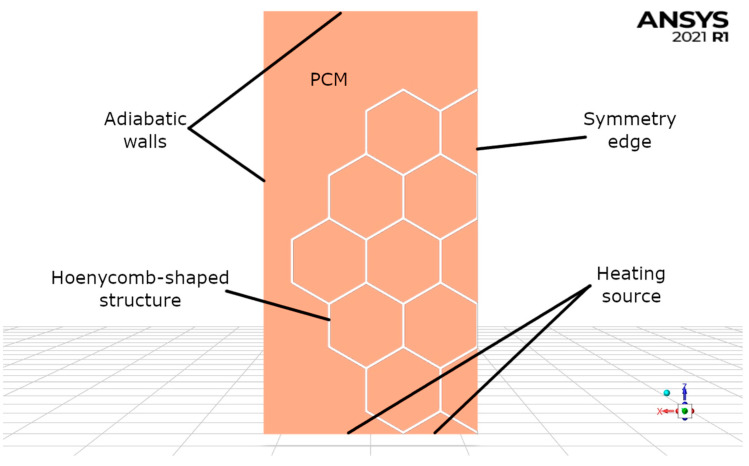
Boundary conditions employed during calculations. Images used courtesy of ANSYS, Inc.

**Figure 4 materials-16-01207-f004:**
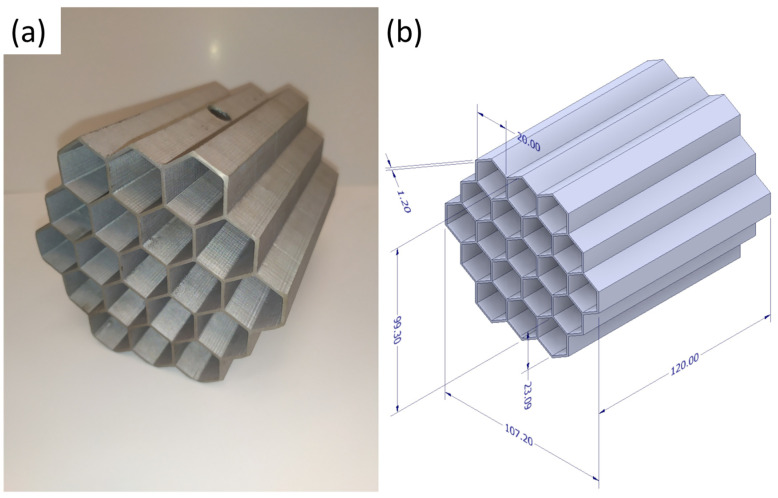
The honeycomb structure: (**a**) cast element made of EN AC 44200 Al-Si alloy; (**b**) dimensions of 1.2 mm wall-thick structure (in experiment and simulation).

**Figure 5 materials-16-01207-f005:**
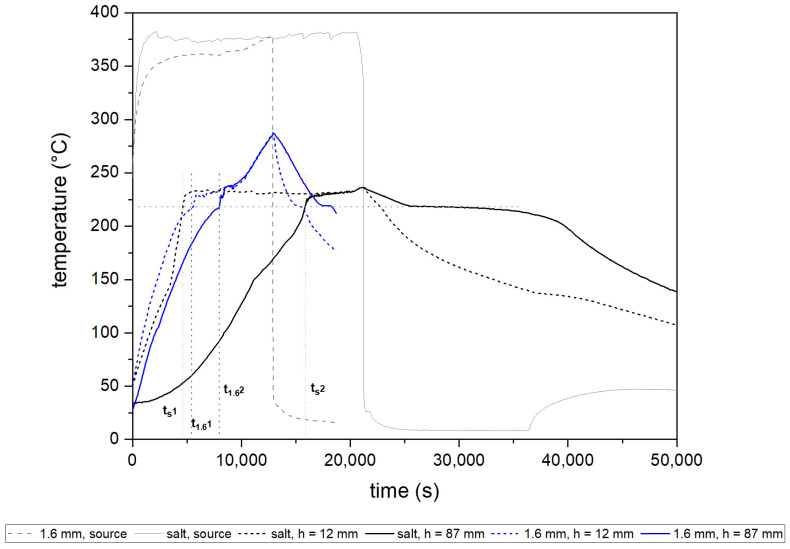
The temperature change in the deposit during the heating and cooling of the store: containing pure PCM and the one with honeycomb heat transfer enhancer with a wall thickness of 1.6 mm.

**Figure 6 materials-16-01207-f006:**
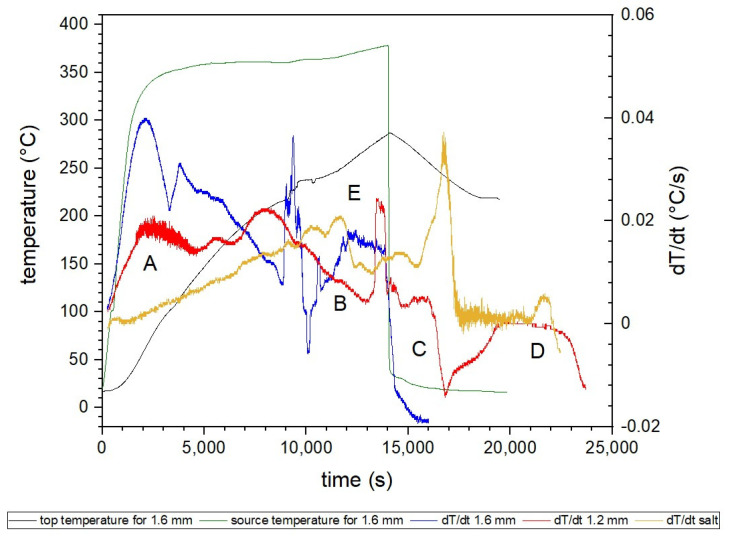
Derivative curves for the investigated cases and the temperature course of the heat source and the top (h = 87 mm) thermocouple for the 1.6 mm insert. Letters symbolize areas as follows: A—the first peak responsible for heat conduction through the insert, B—phase change area, C—the beginning of water cooling, D—crystallization process, E—change of crystalline system of KNO_3_.

**Figure 7 materials-16-01207-f007:**
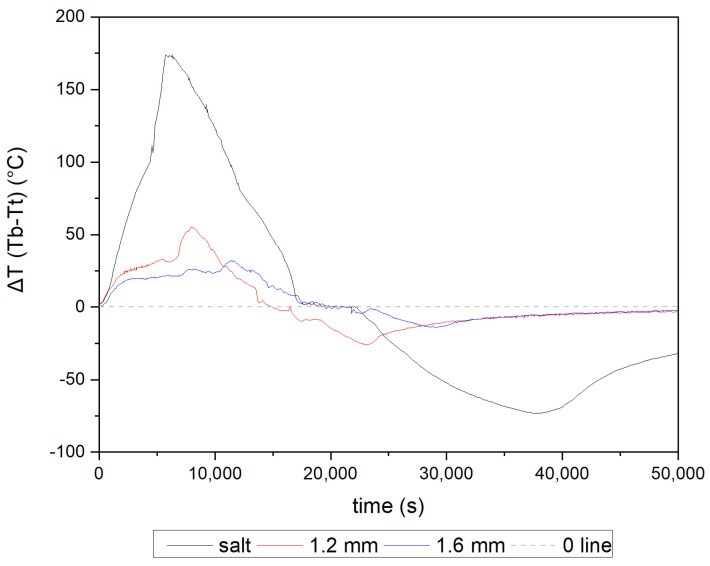
The temperature gradient between the top and bottom thermocouple during the experiment for all tested cases.

**Figure 8 materials-16-01207-f008:**
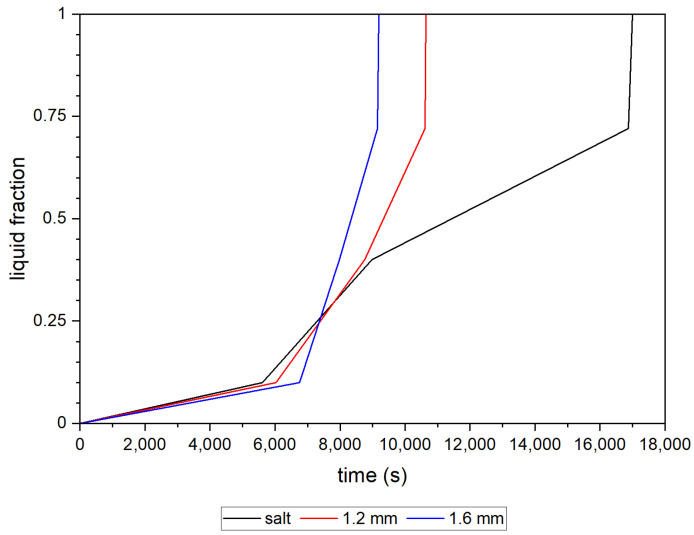
Liquid phase formation estimated on the basis of the excess temperature of the phase transition at the measuring points.

**Figure 9 materials-16-01207-f009:**
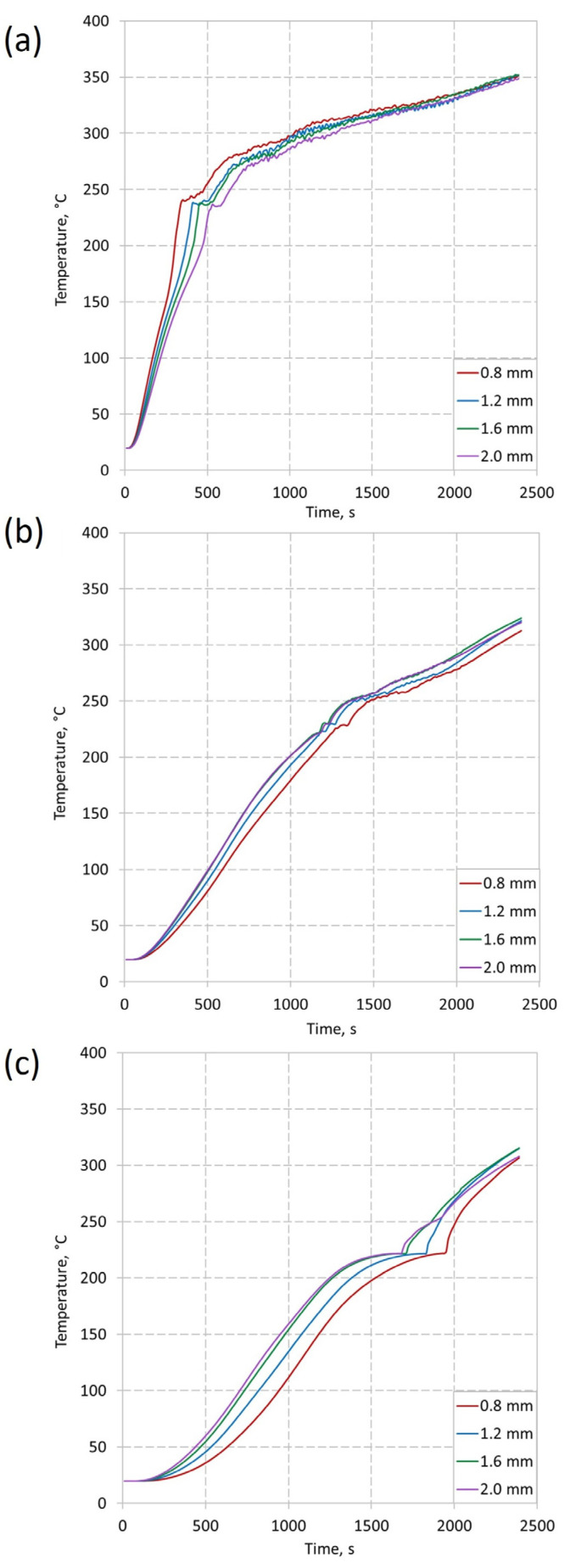
Calculated temperatures evolution on three different heights, for four thicknesses of the insert: (**a**) h = 12 mm; (**b**) h = 50 mm; (**c**) h = 87 mm.

**Figure 10 materials-16-01207-f010:**
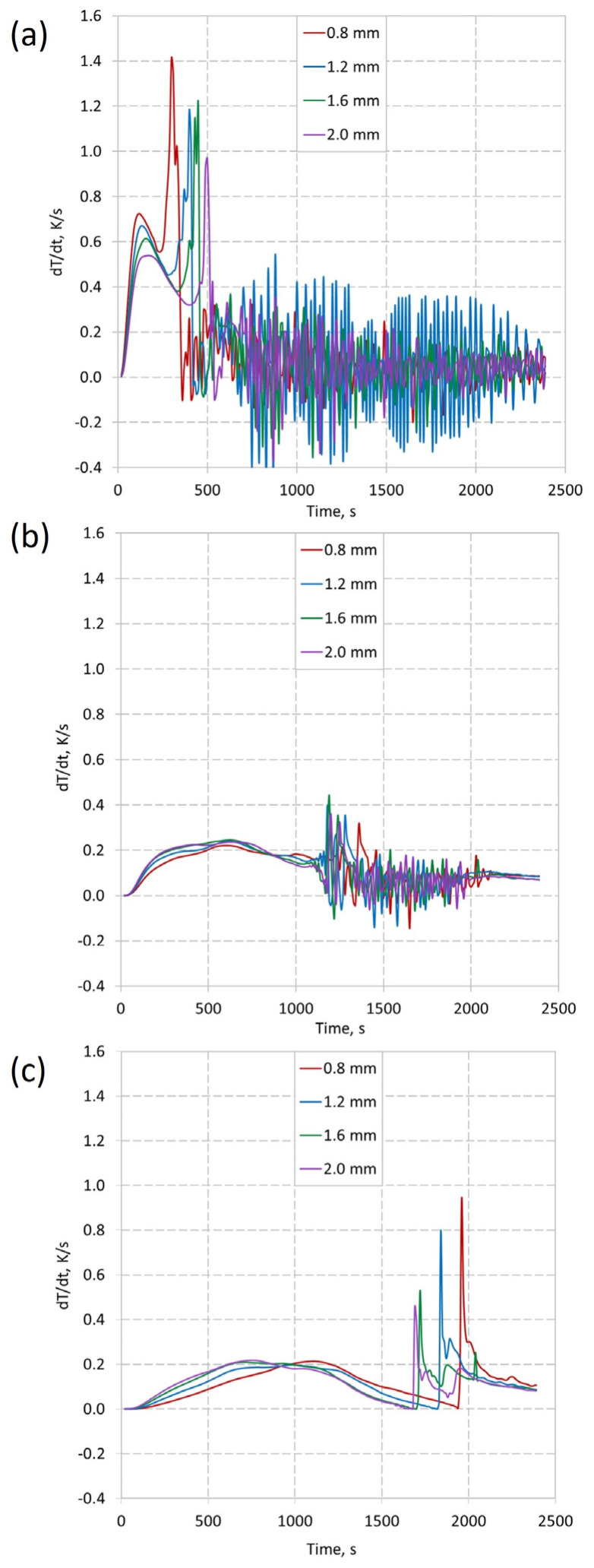
Derivation evolution on three different heights, for four thicknesses of the insert: (**a**) h = 12 mm; (**b**) h = 50 mm; (**c**) h = 87 mm.

**Table 1 materials-16-01207-t001:** Mesh quality in domains.

Domain	Hexahedral Elements	Quality		Thickness of Elements Adjacent	
		Min	Max	Min	Max
1	3168	1	1	0.400148	1.84752
2	3528	0.758	1	0.050000	1.47029
3, 8, 13	243	0.692	0.956	0.133432	1.15556
4–7, 9–12, 14	361	0.95	0.974	0.607737	1.21547

**Table 2 materials-16-01207-t002:** Thermophysical properties of materials used during calculations [26,27,28,29].

Material	Density, kg/m^3^	Specific Heat, J/(kg·K)	Thermal Conductivity, W/(m·K)	Dynamic Viscosity, kg/(m·s)	Melting Heat, J/kg	Melting Temperature, K
PCM	Equation (1)	1492	0.457	0.0063	108,000	495
Aluminum	2644	960	160	-	-	-

**Table 3 materials-16-01207-t003:** Starting time of the deposit’s melting at the height of the bottom (h = 12 mm) thermocouple (t_1_) and the top (h = 87 mm) thermocouple (t_2_) and the calculated duration of the deposit’s melting (delta) for the tested cases: a pure mixture of salts, PCM, and an insert with a wall thickness of 1.2 mm, and a PCM insert with a wall thickness of 1.6 mm.

Time	PCM	1.2 mm	1.6 mm
t_1_ [s]	4710–5590	5480–7540	5220–7920
t_2_ [s]	15,960–16,850	10,570–12,010	7850–10,400
Δ (t_2_ − t_1_) range [s]	10,950–11,900	4640–5700	2420–2620
Δt average [s]	11,425	5170	2520
Δt average [min]	190.42	86.17	42.00

**Table 4 materials-16-01207-t004:** Temperature values for used thermocouples (b—bottom h = 12 mm, c—center h = 50 mm, t—top h = 87 mm) at the time t_1_ and t_2_ (from Table 1) for the application of different enhancers.

Temperature Gradient		PCM	1.2 mm	1.6 mm
t_1_	T_b_ [°C]	218.3	218.8	218
	T_c_ [°C]	98.1	191.5	204
	T_t_ [°C]	53.2	170.8	185.7
t_2_	T_b_ [°C]	230	231.1	232.6
	T_c_ [°C]	229.5	232.3	230.7
	T_t_ [°C]	218.3	218.8	218
ΔT for t_1_		165.1	48	32.3

## Data Availability

Not applicable.

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
