# Peer review of "Optimization of Hexagonal Structure for Enhancing Heat Transfer in Storage System"

_materials, 2023, doi:10.3390/ma16031207_

Round 1

Reviewer 1 Report

This study aims to optimize hexagonal structure for enhancing heat transfer in storage system. Cast aluminum honeycomb-shaped structures were produced via 3D printing of polymer model and investment casting. Only two different wall thicknesses of 1.2 mm and 1.6 mm were tested. More data of different wall thicknesses are required to draw more accurate conclusion. The current form is more about simply stating results, and lacks more in-depth analysis. It is not recommended for publication in its current form.

Author Response

Thank you very much for drawing attention to the need for a broader spectrum of research. Different wall thicknesses were modelled and simulated in ANSYS FLUENT. Afterwards the most promising geometries were fabricated and validated experimentally. Laboratory tests were only carried out for two wall thicknesses of the prepared castings. Charging and discharging cycles were repeated several times, for a several weeks. Authors selected the most representative measurements for article. But indeed, further measurements would better explain some phenomena and draw more accurate conclusion. Making thinner walls, less than 1.2 mm, is possible, the team achieved walls as thin as 0.6 mm, however, the stiffness and strength of such inserts is very low. Tests from previous years in heat accumulators have shown that fatigue failure occurs. On the other hand, making walls thicker than 1.6 mm increases the mass of the system and reduces the fraction of PCM in the chamber. Therefore, following the reviewer's instructions, we will start making inserts with an intermediate wall of 1.4 mm and checking them in the accumulator. The Authors made every effort to improve the quality of the analysis of the results carried out by completing the missing information indicated, inter alia, by the other Reviewers. The time received to apply the corrections, which, in this case, would include subsequently modelling, manufacturing the casting, and conducting a series of charging and discharging tests on the accumulator, is insufficient to carry out a series of such time-consuming and complicated tasks. The Authors hope, that the current version of the Article will at least partially fill the gaps identified.

Please see the attachment for the revised manuscript with marked-up changes in review mode.

Reviewer 2 Report

Thematically the work is interesting for the researchers and professionals and the proposed manuscript is relevant to the scope of the journal.

I found it appropriate for publication in the Materials journal, but only after some modifications and clarification from the Authors.
The title is a clear representation of the manuscript's content. 

The overall organization and structure of the manuscript are appropriate. The paper is well written and the topic is appropriate for the journal.
The aim of the paper is well described and the discussion was well approached, its results and discussion are correlated to the cited literature data.

The literature review is comprehensive and properly done.

The novelty of the work must be more clearly demonstrated.

Statistical interpretation of the analytical data must be more properly presented.

Other Specific Comments: The work is properly presented in terms of the language. The work presented here is very interesting and well done, it is presented in a compact manner.
In general, there are no doubtful or controversial arguments in the manuscript. The methodology applied in the research is presented in clear manner, so that it is repeatable by other authors.
The results are presented in a logical sequence and the discussion and analysis of the results are properly elaborated. 

Perhaps more details could be introduced in section 2.2 numerical analysis, this is the main part of text for readers to validate the model.

The main drawback of the paper i s the extent of novelty, or the main novelty in the present work, compared to the works of other researchers? In my opinion, the authors should put additional effort to demonstrate that the present work gives a substantial contribution in the research area.

Author Response

Thank you very much for the comments and kind review. The responses to each comment requiring corrections in the article are listed in separate sections below.

“Perhaps more details could be introduced in section 2.2 numerical analysis, this is the main part of text for readers to validate the model.” Chapter 2.2 (pages 4 – 6) has been expanded with additional information to better understand the CFD model. Added: information on the quality and density of the numerical grid and domains, boundary conditions were specified, and the assumptions were better described.

“The main drawback of the paper is the extent of novelty, or the main novelty in the present work, compared to the works of other researchers? In my opinion, the authors should put additional effort to demonstrate that the present work gives a substantial contribution in the research area.” – The explanation has been added to the conclusions (pages 16 and 17) as follows: “The main innovation presented in the article is a manufacturing method which allows new thin-walled, fully customized, complex structures with the possibility of rapid modifications to be fabricated. The elements produced via investment casting method are characterized by high precision of execution, repeatability and extensively developed surface area, allowing good contact between PCM and the material of the enhancer. Before that, the enhancers were commonly produced of stainless steel by the means of extrusion, which was a huge limitation when it comes to the use of complex shapes. Moreover, the use of stainless steel in molten salt environment carries a high risk of corrosion of the system, so the service life of the system will be at danger of being drastically reduced. The use of a customized, aluminum enhancer reduces the chances of such phenomena and allows the insert to be optimally adapted to the case study. In addition, the developed method allows to the production of any metal shapes quickly, much cheaper than, e.g. selective laser melting (SLM).”

Please see the attachment for the revised manuscript with marked-up changes in review mode.

Reviewer 3 Report

The paper is well written with sufficient literature review on the topic.  A computational and experimental study has given good insight into the problem. Results and discussions have been properly explained with relevant graphs. However following shortcomings or flaws have been noticed.

1.       Why have the authors taken the thermal conductivity in solid state and the liquid state as same values even though the values are different? What are the values?

2.       What are the boundary conditions chosen for the study?

3.       Have the authors performed the grid independence study? Give the details.

Give the details of the instruments used for the measurement ot temperature-related parameters suchas make, specifications, accuracy etc.

4.       What does the huge negative ΔT for the PCM salt indicate in fig 6?

5.       Figure 8 line thickness can be increased to have more clarity and visibility.

6.       Fig 9 is very confusing. More discussion is needed on its the characteristics.

7.       Conclusions can be given as bulleted points

Author Response

Thank you for all valuable comments. In the next part of the response, the answers will be given accordingly to each point stated by the Reviewer.

AD1. The material data was collected from the papers. Thanks to the Reviewer's attention, they have been just added to the references in Table 2. The papers are cited as follows in the “References” point (page 18):

  1. Bauer, T.; Laing, D.; Tamme, R. Overview of PCMs for Concentrated Solar Power in the Temperature Range 200 to 350°C. In Proceedings of the AST; October 27 2010; pp. 272–277.
  2. D’Aguanno, B.; Karthik, M.; Grace, A.N.; Floris, A. Thermostatic Properties of Nitrate Molten Salts and Their Solar and Eutectic Mixtures. Sci Rep 2018, 8, 10485, doi:10.1038/s41598-018-28641-1.
  3. Kamimoto, M.Y. Thermodynamic Properties of 50 Mole % NaNO3 - 50% KNO3 (HTS2). Thermochim Acta 1981, 48, 319–331.
  4. Zhao, Q.-G.; Hu, C.-X.; Liu, S.-J.; Guo, H.; Wu, Y.-T. The Thermal Conductivity of Molten NaNO3, KNO3, and Their Mixtures. Energy Procedia 2017, 143, 774–779, doi:10.1016/j.egypro.2017.12.761.

Unfortunately, the values collected were describing salt mixture only in liquid state, that is why the Authors decided to apply a simplification using values they know, rater than approximate and calculate potential unknown value.

AD2. In fact, the boundary conditions have not been described in detail. Therefore, in the new version, in chapter 2.2 (pages 4 – 6), explanations, a table, a drawing and assumptions have been added.

AD3. The information about the utensils used to perform thermal cycling process, as well as to collect and record temperature data was added to the article as follows: ”The deposit was heated up with the use of a heating plate (Hot Plate, Model SH-II-5B, heating power 1200 W, temperature accuracy ±1%. The experimental data, temperature at different heights in the bed, was registered by K-type NiCr-NiAl thermocouples, class 1 (±1.5 °C) according to the PN-EN 60584 standard, collected and recorded by Adam 4018 type adapter with 8-channels, and Visidaq program.”, and can be seen on page 3.

The authors performed only a few mesh density tests. However, due to the longtime of individual calculations (nearly 2 weeks), they were not performed for the full cycle. They were used to identify and eliminate sensitive computing areas. This resulted in mesh densification in areas of lower mesh quality, as described in section 2.2. Thanks to these quick tests, some areas of the mesh were densified to as little as 0.05 mm. It is also worth noting that the main aspect related to heat transfer in the insert is not a computational problem. Therefore, due to a different purpose of the work, the grid independence study was not subject to a broader analysis in this work.

AD4. Thank you for pointing out this interesting aspect, which was unfortunately overlooked by the Authors. A paragraph describing and explaining this phenomenon was added to the page 10, as follows: “The huge negative ΔT visible for only PCM deposit is strictly connected to the heat transfer rate in the bed. Salt mixture in solid state is characterized by low thermal conductivity (for KNO3-NaNO3 mixture 0.457 W/(m·K)), thus even if the part of the deposit is already melted, the heat provided to the rest of the material is significantly lower than the amount enabling phase change. In case of enhanced structures, the heat is transported through the aluminum structure, a material with considerably higher thermal conductivity of 160 W/(m·K), therefore the melting process, as well as strictly connected to the material temperature, is more evenly distributed in the system”.

AD5. Quality and clarity of Fig.9 and Fig.10 have been improved.

AD6. In Chapter 3.2, a commentary to Fig.10 was added, explaining the authors' observations in detail (page 16), as follows: “As shown in Figure 10, the heat transport rate is very high at the beginning of the process, causing large changes in temperature in time, mainly at the height of h = 12 mm, where the phase change begins the fastest. The temperature gradient is the highest among the 3 analyzed heights and exceeds 0.5 K/s. The rapidly increasing temperature over time is the result of a short distance from the heat source. Moreover, the highest temperature increase at h = 12 mm was obtained for the insert thickness of 0.8 mm. This may be due to slower heat transfer in the vertical direction. After this stage, there is a noticeable, especially in Figure 10b, decrease in dynamics. It is caused by reaching the temperature at which the phase change takes place. At h = 12 mm, the dynamics decrease but increase at other heights. It is worth noting that after the de-scribed phase change, the dT/dt values oscillate close to zero. This oscillation is also visible in Figure 10b, after the phase change. This may be due to the convective motion of the PCM and the sinking of the solid phase. This would correspond to the non-uniform temperature changes visible, especially in Figure 9a. The influence of the wall thickness at the height of h = 50 mm and h = 87 mm is visible, where the dynamics increase with increasing wall thickness. This confirms the observations described for h = 12 mm, for which the phase change occurs later, with the increase in wall thickness. This suggests increased heat transfer along the insert to the top of the vessel. Figure 10b shows the effect of using a thicker insert. The phase change process is the slowest there for a wall thickness of 0.8 mm. However, for the other thicknesses, the phase change time is similar. Figure 10c shows the best effect of using inserts with a larger wall thickness. The phase change process at the height of h = 87 mm (Figure 10c) occurs the fastest for a wall thickness of 2.0 mm. It should be noted that the rate of temperature change in time does not increase linearly with increasing wall thickness. It is also visible in the maximum values of dT/dt, i.e. right after the PCM has been melted. For a wall thickness of 2.0 mm, there is the smallest increase in dT/dt during the phase change (shown in Figure 9c). For thicknesses 0.8, 1.2, 1.6 and 2.0, the maximum values were 0.94, 0.80, 0.52 and 0.45, respectively. This suggests a more even heat distribution through the insert. This also reduces the temperature gradient throughout the vessel, which is a desirable effect”.

AD7. Definitely, presenting the conclusions of the work in the form of bullet points highlighting the most important information will be the clearest form of dissemination. Thank you for drawing attention to this. The conclusions have been slightly shorten and stated as follows: ”Based on the research and results presented, the conclusions can be drawn as follows:

  • Additive manufacturing followed by investment casting allowed for obtaining customized complex metal structures.
  • The use of heat enhancer can notably increase the rate of charging. In the case of studied systems, melting temperature was reached two times faster in case of the system enhanced with a 1.2 mm–thick structure in comparison to the pure deposit of salt PCM. The presence of the casting influences the behavior of the heat source.
  • Due to the nonconductive properties od used salt mixture, the application of the insert with 1.2 mm walls improved the time of charging twice, while 1.6 mm – four times. Additionally, the use of the thicker insert lowered the temperature gradient at the beginning of phase change in the deposit 5 times, compared to pure PCM.
  • Even for salt mixture, phase change from rhombic to triagonal structure of KNO3 in the temperature range of 130 – 150°C is observed in the form of flattering the temperature-in-time derivative plot.
  • According to the numerical analysis, which confirmed the beneficial effect of using inserts with higher wall thicknesses, it can be concluded that the higher thickness of the enhancer wall, the better distribution of heat throughout the vessel volume is observed.
  • The main innovation presented in the article is a manufacturing method which allows new thin-walled, fully customized, complex structures with the possibility of rapid modifications to be fabricated. The elements produced via investment casting method are characterized by high precision of execution, repeatability and extensively developed surface area, allowing good contact between PCM and the material of the enhancer. Before that, the enhancers were commonly produced of stainless steel by the means of extrusion, which was a huge limitation when it comes to the use of complex shapes. Moreover, the use of stainless steel in molten salt environment carries a high risk of corrosion of the system, so the service life of the system will be at danger of being drastically reduced. The use of a customized, aluminum enhancer reduces the chances of such phenomena and allows the insert to be optimally adapted to the case study. In addition, the developed method allows to the production of any metal shapes quickly, much cheaper than, e.g. selective laser melting (SLM).”

The text is placed on pages 16 and 17.

Please see the attachment for the revised manuscript with marked-up changes in review mode.

Round 2

Reviewer 3 Report

The authors have satisfactorily answered all the queries raised and also have included them in the manuscript. Paper can be accepted.